# Building Trust in Decision with Conformalized Multi-view Deep Classification

Wei Liu
College of Electronic and Information
Engineering,
Tongji University,
Shanghai, China
ldachuan@outlook.com

Yufei Chen*
College of Electronic and Information
Engineering
Tongji University
Shanghai, China
yufeichen@tongji.edu.cn

Xiaodong Yue
Artificial Intelligence Institute,
Shanghai University,
Shanghai, China
yswantfly@shu.edu.cn

## Abstract

Uncertainty-aware multi-view deep classification methods have markedly improved the reliability of results amidst the challenges posed by noisy multi-view data, primarily by quantifying the uncertainty of predictions. Despite their efficacy, these methods encounter limitations in real-world applications: 1) They are limited to providing a single class prediction per instance, which can lead to inaccuracies when dealing with samples that are difficult to classify due to inconsistencies across multiple views. 2) While these methods offer a quantification of prediction uncertainty, the magnitude of such uncertainty often varies with different datasets, leading to confusion among decision-makers due to the lack of a standardized measure for uncertainty intensity. To address these issues, we introduce Conformalized Multi-view Deep Classification (CMDC), a novel method that generates set-valued rather than single-valued predictions and integrates uncertain predictions as an explicit class category. Through end-to-end training, CMDC minimizes the size of prediction sets while guaranteeing that the set-valued predictions contain the true label with a user-defined probability, building trust in decision-making. The superiority of CMDC is validated through comprehensive theoretical analysis and empirical experiments on various multi-view datasets.

## CCS Concepts

• **Computing methodologies** → **Supervised learning by classification**; **Neural networks**.

## Keywords

Multi-view classification, uncertainty estimation, conformal prediction.

**ACM Reference Format:**
Wei Liu, Yufei Chen, and Xiaodong Yue. 2024. Building Trust in Decision with Conformalized Multi-view Deep Classification. In *Proceedings of the*

*Corresponding author.

*32nd ACM International Conference on Multimedia (MM '24), October 28-November 1, 2024, Melbourne, VIC, Australia.* ACM, New York, NY, USA, 10 pages. https://doi.org/10.1145/3664647.3681297

## 1 Introduction

In practical environments encompassing image analysis, computer vision, data mining and multimedia applications, objects are frequently characterized by numerous modalities or feature varieties, jointly categorized as multi-view data [45]. This phenomenon has engendered the evolution of multi-view learning, a field that strives to construct superior methods for integrating multiple views to augment performance. Leveraging the achievements of deep learning, Multi-view Deep Learning (MDL) has emerged as a significant research trajectory aimed at acquiring shared multi-view representations through Deep Neural Networks (DNNs) [2, 20, 38, 42].

While multi-view deep learning demonstrates notable effectiveness in real-world scenarios, such developments have largely thrived within the *closed-world assumption*, which assumes an ideal scenario where data views are error-free and training aligns perfectly with testing distributions. However, the complex realities of the real world often challenge this idealized assumption, especially with the presence of noise-corrupted multi-view data, commonly referred to as outliers, which introduce risks during the testing phase. In response to this challenge, recent research has introduced uncertainty-aware multi-view deep classification methods [15, 16, 28, 29, 50, 51] that aim to assess uncertainty levels in predictions to enhance the reliability of multi-view results. Despite their evident value, these approaches encounter limitations that may elevate risks in real-world applications, particularly those of high stakes (e.g., medical diagnosis) that require critical decision-making.

Specifically, imagine you're a doctor facing a critical medical decision, such as diagnosing pancreatic cancer, based on diagnostic information from a multi-view deep classifier. In scenarios where medical imaging is challenging to classify due to inconsistencies among different views or when encountering images of rare diseases not present in the training data, what output would you expect from the classifier to make the best decision? Existing uncertainty-aware multi-view methods typically produce a single-valued prediction with a degree of uncertainty. However, in such scenarios: **1)** Information gathered from inconsistent views often leads to conflicting evidence among different classes (i.e., the Normal class and Serous Cystic Neoplasm (SCN) class as illustrated in Figure 1(a)), making it challenging to reach a conclusive decision. **2)** Although the uncertainty density distribution of rare diseases may differ from that of the original imaging, determining whether an image belongs

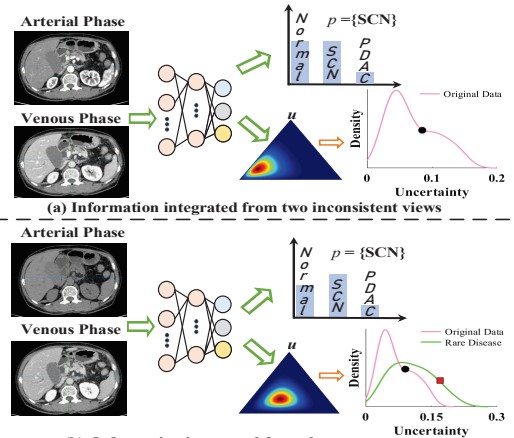

**Figure 1: The output format of existing uncertainty-aware multi-view deep classification methods for pancreatic cancer diagnosis presents a class probability map alongside estimated uncertainty degrees. The classes include "Normal", "SCN", and "PDA". Here, $p$ denotes the prediction result, while $u$ indicates the uncertainty degree. Pink and green lines represent the uncertainty density distribution derived from original medical imaging and rare disease, respectively. The black circle denotes the position of the left sample in (a) within the uncertainty density distribution of the original data, while the red square indicates the position of the left sample in (b) within the uncertainty density distribution of rare diseases.**

to a rare disease, referred to as outliers in machine learning, can be difficult due to the small difference between outlier and normal data with a small magnitude (valued range from $[0 - 0.3]$), as illustrated in Figure 1(b). In contrast, the desired output should: **1)** Provide possible prediction sets instead of deterministic single-valued predictions, allowing for the ruling in or ruling out of harmful decisions (such as diagnosing Pancreatic Ductal Adenocarcinoma (PDAC), a malignant tumor category). **2)** Have the ability to make a clear judgment as to whether the samples belong to outliers in real-world environments across different datasets. **3)** Provide a theoretical guarantee of coverage, ensuring that the true label is included within the prediction sets with a high probability.

To this end, we introduce a novel Conformalized Multi-view Deep Classification (CMDC) method. It is designed to offer prediction sets, which are subsets of candidate labels explicitly including outlier classes for classification tasks, thereby enhancing decision-making reliability. Drawing inspiration from the Evidential Neural Network (ENN) [34] and the theory of Conformal Prediction (CP) [3], our model initially captures the second-order probability of the class distribution to explicitly estimate prediction uncertainty caused by outliers, thus expanding the output space by incorporating outlier classes. Leveraging an effective multi-view fusion strategy and the conformal score function, we measure the similarity between calibration samples and testing inputs, enabling the construction of suitable prediction sets. Through end-to-end training, our method minimizes the size of prediction sets while

providing a coverage guarantee that the set-valued predictions contain the true label with a user-defined probability, thereby building trust in decision-making. In summary, the specific contributions of this paper are as follows:

(1) We introduce a novel multi-view model aimed at providing reliable prediction sets, which are subsets of candidate labels that explicitly include outlier class, thereby enhancing the reliability of decisions. This approach marks a new paradigm in multi-view deep classification.

(2) The proposed model presents a unified framework designed to ensure that the true label is contained within the prediction sets. This is achieved through the incorporation of a suitable multi-view fusion strategy and conformal score function, both of which are implemented in an optimizable (learnable) manner.

(3) Theoretical analysis demonstrates that our model effectively decreases the empirical risk of learning outcomes as the number of views increases, while also controlling the conformal risk within a small margin.

## 2 Related Works

### 2.1 Multi-view Deep Learning

In recent years, numerous multi-view deep learning methods have emerged, integrating multi-view learning with DNNs to achieve high performance [26, 27, 46, 48, 49]. Traditional methods include Canonical Correlation Analysis (CCA) combined with DNNs [2, 42, 43]. CCA aims to maximize the correlation between different views to find a common representation. Deep CCA (DCCA) [2] focuses on capturing nonlinear relationships, while Deep Canonically Correlated AutoEncoder (DCCAE) [42] trains autoencoders to obtain common representations. Deep Tensor Canonically Correlation Analysis (DTCCA) [43] learns complex nonlinear transformations of multiple views (more than two) of data to ensure that the resulting representations are linearly correlated in high order.

Recently, several uncertainty-based multi-view classification methods [15, 16, 28, 29, 44, 50, 51] have been proposed. Han et al. [15, 16] focus on uncertainty estimation using Dempster's rule of combination to produce reliable results, while Liu et al. [29] treat the fusion of multiple views as an opinion aggregation process. Despite their success in multi-view deep classification, these methods fall short in addressing the reliability of decisions in real-world environments. One limitation is their restriction to providing a single class prediction per instance, which may lead to inaccuracies when dealing with samples that are difficult to classify due to inconsistencies across multiple views. Additionally, while these methods quantify prediction uncertainty, its magnitude often varies with different datasets, causing confusion among decision-makers due to the lack of a standardized measure for uncertainty intensity. *In contrast, our method generates set-valued predictions instead of single-valued ones and integrates uncertain predictions as an explicit class category, thereby enabling more reliable decisions.*

### 2.2 Uncertainty Estimation in Deep Learning

Uncertainty estimation is pivotal for ensuring trustworthy decision-making and has been extensively explored in deep learning [1, 11]. Two primary types of uncertainties, namely aleatoric uncertainty

and epistemic uncertainty, serve as indicators of the likelihood that predictions given by models are incorrect. Uncertainty-based deep learning models can generally be classified into Bayesian and non-Bayesian models. Bayesian deep learning approaches [18] facilitate the interpretation of model parameters by replacing deterministic weight parameters with distributions. These methods employ various approximation techniques, such as Monte Carlo (MC) dropout [12], Markov Chain Monte Carlo (MCMC) [33], Variational Inference (VI) [39], and Laplacian approximations [31], to estimate uncertainty. However, Bayesian methods often entail high computational costs. To address this, several non-Bayesian methods have been proposed. For instance, Deep Ensemble [19, 24] combines multiple deep neural networks to estimate predictive uncertainty. More recently, another approach termed as ENN [34], directly models uncertainty using subjective logic [21] and evidence theory [7, 35] instead of indirectly through weights. Simultaneously, several uncertainty estimation algorithms based on the Dirichlet distribution have been introduced [30, 34], aiming to estimate uncertainty in distribution shifts. *Given the robustness of subjective logic and evidence theory in estimating uncertainty caused by outliers without additional computational burden, we employ the Evidential Neural Network (ENN) as the backbone of our uncertainty-aware module.*

## 2.3 Conformal Prediction

Conformal Prediction (CP) serves as an uncertainty quantification tool in machine learning, aimed at enhancing the reliability of models [3, 8, 36, 37]. At its core, CP involves a statistical calibration technique for predictive models, offering finite-sample guarantees on the predicted uncertainty. By specifying a permissible level of risk, conformal calibration adjusts the model's predictions into predictive sets or intervals. These sets are designed to contain the true test labels or targets at least as frequently as dictated by the specified risk level. There are two primary steps in CP. Firstly, in the prediction step, a trained model computes conformity scores, which gauge the similarity between calibration examples and a testing input. Secondly, in the calibration step, these conformity scores from a set of calibration examples are utilized to determine a threshold for constructing prediction sets that adhere to the coverage constraint (e.g., $1 - \tau = 90\%$). Recent studies have proposed conformity scores based on ordered probabilities in CP for single-view classification [4, 32]. *However, there has been limited focus on multi-view classification within an end-to-end training setup, especially regarding the integration of the outlier class into the prediction set through the development of an effective conformal score function. This gap in the literature motivates the development of the work presented in this manuscript.*

## 3 Method

In this section, we start with notations and problem statement, followed by detailing the proposed CMDC and its key components.

### 3.1 Problem Formulation

Considering a multiclass classification task on a multi-view dataset $\mathcal{X} = \left\{ \left\{ x_i^v \right\}_{v=1}^{V}, y_i \right\}_{i=1}^{N}$ with $V$ views, where $x_i^v \in \mathcal{X}$ denotes the $v^{th}$ view of the $i^{th}$ sample, and $y_i \in \mathcal{Y} = \{1, \ldots, K\}$ represents the

associated class label. Our goal is to learn a conformalized multi-view model $h\left( \left\{ x_i^v \right\}_{v=1}^{V}; \theta \right)$ that maps $\left\{ x_i^v \right\}_{v=1}^{V}$ to a prediction set $C_\theta \left( \left\{ x_i^v \right\}_{v=1}^{V} \right) \subseteq 2^{K+1}$ containing the true label $y_i$ with a coverage probability of $1 - \tau$, where the $(K+1)^{th}$ class represents the outlier category. It's evident that achieving a large coverage with a small set size will result in better prediction performance.

To realize this goal, we need to address three key challenges: **1)** Explicitly estimating prediction uncertainty caused by outliers in multi-view setting; **2)** Developing an effective fusion strategy to enhance performance, including classification accuracy and prediction set size, through the fusion of multiple views; **3)** Guaranteeing that prediction sets contain the true label and correctly identify outlier classes with a user-defined probability (*i.e.*, $1 - \tau = 90\%$).

To this end, we draw inspiration from both the ENN and CP and propose a novel conformalized multi-view deep classification model. This model not only ensures the effective fusion of multiple views but also provides prediction sets with guaranteed coverage, thereby enhancing the precision and reliability of classification results in the face of challenges posed by inconsistent views or outliers.

### 3.2 Augmented Output Space in CMDC

To incorporate the outlier category into the possible label sets, providing intuitive assessment to decision-makers, we expand the multi-view output space $y_i \subseteq 2^K$ to $y_i \subseteq 2^{K+1}$. To achieve this, two aspects need assurance: 1) A reliable tool to identify outliers in the input space; 2) Ensuring that the probabilities of the outlier class and the other normal classes sum to one. To this end, we first assess the quality of data views through uncertainty estimation in ENN, a common practice in existing state-of-the-art works [15, 16, 28, 29].

Formally, let $h_v \left( x_i^v; \theta \right)$ be the output of the penultimate layer (i.e., logits layer) of a neural network for the sample $x_i^v$ in single view. Instead of using *softmax* to predict a single categorical estimation, we use another activation function $a(\cdot)$ (i.e., ReLU) to capture a non-negative evidence $e_i^v = a\left( h_v \left( x_i^v; \theta \right) \right)$, $e_i^v = \left[ e_{i1}^v, \ldots, e_{iK}^v \right]$, over $K$ categories. Then we have:

$$b_{ik}^v = e_{ik}^v \Big/ S_i^v,$$

$$u_i^v = 1 - \sum_{k=1}^{K} b_{ik}^v = K \Big/ S_i^v, \tag{1}$$

where $b_{ik}^v$ is the belief mass of the $k^{th}$ class based on the collected evidence, $u_i^v$ is the estimated prediction uncertainty that indicates the lack of total evidence, and $S_i^v = \sum\limits_{k=1}^{K} e_{ik}^v + K$. When the evidence is low, leading to a low belief mass $b_{ik}^v$ and high uncertainty $u_i^v$ in prediction. Conversely, when the evidence is high and concentrated in one category, the uncertainty will be low.

According to the Eq.(1), we can incorporate the outlier class into augmented single-view output space $O_i^v = \left\{ b_i^v = \left\{ b_{ik}^v \right\}_{k=1}^{K+1} \right\}$, where $b_{i(K+1)}^v = u_i^v$. Then, we expand it to multi-view output space $O_i = \left\{ b_i = \{ b_{ik} \}_{k=1}^{K+1} \right\}$ through proposing an effective multi-view fusion strategy as follows:

**Definition 1.** *(Fusion strategy across multiple views).* Given the output space of two views $O_i^1$ and $O_i^2$, the fusion $O_i = O_i^1 \oplus O_i^2$ can be achieved by the following rule:

$$b_{ik,k \neq K+1} = \frac{b_{ik,k \neq K+1}^1 b_{i(K+1)}^2 + b_{ik,k \neq K+1}^2 b_{i(K+1)}^1}{b_{i(K+1)}^1 + b_{i(K+1)}^2 - b_{i(K+1)}^1 b_{i(K+1)}^2},$$

$$b_{i(K+1)} = \frac{b_{i(K+1)}^1 b_{i(K+1)}^2}{b_{i(K+1)}^1 + b_{i(K+1)}^2 - b_{i(K+1)}^1 b_{i(K+1)}^2}. \tag{2}$$

Without loss of generality, let $O_i = \left\{ b_i = \{b_{ik}\}_{k=1}^{K+1} \right\}$ be multi-view output space fused from $V$ views, we can easily expand the above rule to multiple views fusion:

$$O_i = \oplus_{v=1}^V O_i^v = O_i^1 \oplus \cdots \oplus O_i^V. \tag{3}$$

Following the above definition, we also can derive the multi-view evidence $e_i = [e_{i1}, \ldots, e_{iK}]$, over $K$ categories using the Eq.(1).

## 3.3 Conformalized Multi-view Predictor

According to the above augmented multi-view output space, we will introduce how to formalize the conformalized multi-view predictor to provide reliable prediction sets.

Firstly, we define the prediction set (on test data) $C_\theta \left( \{x_i^v\}_{v=1}^V \right)$, which depends on the model parameters $\theta$, across the augmented multi-view output space $O_i$. Specifically, the construction of the prediction sets can be dived into two parts: **1)** The inclusion of the predicted label sets $\hat{y}_i \subseteq 2^k, k \in [1, \ldots, K]$ with the supervision of the true label; **2)** The inclusion of the predicted label $\hat{y}_i = \{K+1\}$ to indicate when test points do not come from the same distribution.

To achieve this, we develop a conformalized multi-view predictor based on the designed conformal score function $E_\theta \left( \{x_i^v\}_{v=1}^V, k \right)$ to obtain the prediction sets. Specifically, following the theory of CP, let $\mathcal{X}_{cal} = \left\{ \{x_i^v\}_{v=1}^V, y_i \right\}_{i=1}^M$ denote a clean calibration dataset from the same distribution as the training dataset, and let $x_i^V$ replace $\{x_i^v\}_{v=1}^V$ for convenience. Then, the conformal score function $E_\theta \left( x_i^V, k \right)$ is designed as follows:

$$E_\theta \left( x_i^V, k \right) := \begin{cases} \sum_{j=1}^{k'} b_{i\pi_j(x_i^V)}, & \pi_{k'} \left( x_i^V \right) = k; \quad k \neq \{K+1\} \\ b_{ik}; & k = \{K+1\} \end{cases}. \tag{4}$$

Here, $\pi \left( x_i^V \right)$ is the permutation of $\{1, \ldots, K\}$ that sorts $b_{ik,k \neq K+1}$ from most likely to least likely. When $k = 3$, $\pi_{k'} \left( x_i^V \right)$ means the index of the $3^{th}$ class in the ordered $b_{ik,k \neq K+1}$. $E_\theta \left( x_i^V, k \right)$ indicates the sum of $b_{ik,k \neq K+1}$ from the most likely to the $\left( \pi_{k'} \left( x_i^V \right) \right)^{th}$ likely class. Then, we have the conformal prediction set $C_\theta \left( \{x_i^v\}_{v=1}^V \right) = \left\{ k : E_\theta \left( x_i^V, k \in [1, \ldots, K] \right) \leqslant \hat{p} \right\} \cup \left\{ \{K+1\} : E_\theta \left( x_i^V, k = \{K+1\} \right) > \hat{q} \right\}$.

The threshold $\hat{p}$ and $\hat{q}$ are determined during calibration on $\mathcal{X}_{cal}$. Specifically, $\hat{p}$ is the $\left\lceil \frac{(1-\tau)(1+M)}{M} \right\rceil$-quantile of the ordered

$\left\{ E_\theta \left( x_i^V, y_i \right) \right\}_{i=1}^M$ with true label $y_i \in [1, \ldots, K]$ and the threshold $\hat{q}$ is the $\left\lceil \frac{(1-\tau)(1+M)}{M} \right\rceil$-quantile of the ordered $\{b_{i(K+1)}\}_{i=1}^M$. Then, given a test sample $\{x_{test}^v\}_{v=1}^V$, the calculation of $\hat{p}$ should ensure the following property:

$$\mathbb{P} \left( y_{test}^{true} \in C_\theta \left( \{x_{test}^v\}_{v=1}^V \right) \right) \geq 1 - \tau. \tag{5}$$

And the calculation of $\hat{q}$ should ensure the following property for the predicted label $\hat{y}_{test}$:

$$\mathbb{P} \left( \hat{y}_{test} = \{K+1\} \right) < \tau. \tag{6}$$

**Remark 1.** *Notably, in contrast to the inclusion of the predicted label sets $\hat{y}_i \subseteq 2^k, k \in [1, \ldots, K]$ entailing true labels for supervising calibration on held-out dataset, the inclusion of $\hat{y}_i = \{K+1\}$ introduces an unsupervised dimension due to the lack of true outlier labels. Thus we should guarantee it does not return too many false positives. To this end, we consider it as a hypothesis testing problem. Points that are rejected as outliers have a p-value less than $\tau$ for the null hypothesis of exchangeability with the calibration data. Then we have the Eq.(5) and Eq.(6) to guarantee the method have a suitable error control as shown in Proposition 2.*

## 3.4 Learning to Form Reliable Prediction Sets

In this section, we outline our approach to learning a model that produces reliable prediction sets in a multi-view setting. To enhance clarity, we structure the learning process as a multi-task learning problem, comprising three distinct objectives:
*(1) Maximizing the integration of joint information from multiple views to support correct class identification.*
*(2) Minimizing information while increasing uncertainty when predictions are incorrect.*
*(3) Minimizing the size of the prediction sets.*

At a high level, the first part (1) involves fitting our multi-view data to the proposed model, thereby enhancing its ability to capture the information for accurate predictions. The second part (2) enforces a prior to eliminate incorrect information, thereby calibrating the estimated confidence to obtain a well-calibrated model. The third part (3) aims to optimize the efficiency of prediction sets. Notably, we do not incorporate a coverage loss as the proposed model theoretically ensures that the true label is contained in prediction sets with a probability of $1 - \tau$.

In summary, for the model $h \left( \{x_i^v\}_{v=1}^V; \theta \right)$, the overall objective function is formulated as follows:

$$\mathcal{L}_{overall}(\theta) = \sum_{i=1}^N \left( \ell_{acc}(\theta)_i + \ell_{cal}(\theta)_i + \ell_{size}(\theta)_i \right), \tag{7}$$

where $\ell_{acc}(\theta)_i$ represents the classification loss term, $\ell_{cal}(\theta)_i$ is the confidence calibration loss term and $\ell_{size}(\theta)_i$ is the set size regularization term.

Specifically, for a training sample $\{x_i^v\}_{v=1}^V$ with one-hot label $y_i$, we treat Dirichlet distribution $Dir(p_i | e_i + 1)$ as a prior on the

likelihood, obtain the negative log likelihood loss:

$$
\ell_{acc}(\theta)_i = \mathbb{E}_{\boldsymbol{p}_i \sim Dir(\boldsymbol{p}_i | \boldsymbol{e}_i + 1)} \left[ -\sum_{k=1}^{K} \log \left( p_{ik}^{y_{ik}} \right) \right]
$$

$$
= \sum_{k=1}^{K} y_{ik} \left( log(S_i) - log(e_{ik} + 1) \right),
$$

(8)

where $S_i = \sum_{k=1}^{K} e_{ik} + K$. Notably, due to the absence of a true outlier label, our focus in this part is solely on the classification of the remaining classes.

To enhance the ability of uncertainty estimation, we propose a confidence calibration loss that regularizes training by applying a confidence prior $c_i = 1 - u_i$ as a penalty, aiming to minimize information for inaccurate predictions:

$$
\ell_{cal}(\theta)_i = \lambda_{t_1} |\boldsymbol{y}_i - \boldsymbol{p}_i| \cdot c_i.
$$

(9)

This loss intuitively imposes a penalty whenever there is an error in the prediction and scales with the confidence of our inferred posterior, which shrinks the evidence for incorrect classes to 0. Larger confidence values are not penalized as long as the prediction is close to the target.

Then, for minimizing the size of prediction sets, we introduce a set size regularization term:

$$
\ell_{size}(\theta)_i = \lambda_{t_2} \sum_{k=1}^{K} \mathbb{1} \left[ E_\theta \left( \{x_i^v\}_{v=1}^{V}, k \right) \leqslant \hat{p} \right].
$$

(10)

To avoid the indicator functions, we use approximation [6] by the sigmoid function $\sigma(x) = 1/1 + \exp(-x)$ to make the set size regularization term differentiable:

$$
\ell_{size}(\theta)_i = \max \left( 0, \lambda_{t_2} \sum_{k=1}^{K} \sigma \left( \hat{p} - E_\theta \left( \{x_i^v\}_{v=1}^{V}, k \right) \right) - \gamma \right).
$$

(11)

Here, we set the $\gamma = 1$ in order to not penalize singletons. Furthermore, we use $\lambda_{t_{1,2}} = \lambda_0 \exp \{-(\ln \lambda_0/T) t\}$ as an annealing factor, where $\lambda_0$ is a small positive constant, $t$ is the current number of training epoch, and $T$ is the total number of training epochs. As the $t$ increasing to $T$, $\lambda_t$ will be exponentially from $\lambda_0$ to 1.

## 4 Theoretical Analysis

In this part, we theoretically prove the effectiveness of our model. We begin by stating a proposition indicating that integrating more views leads to lower empirical risk compared to using fewer views.

**Proposition 1 (Classification Performance Improvement).** *Let* $\widehat{R}\left( h\left( \{x_i^v\}_{v=1}^{\cdot}; \theta \right) \right) = \frac{1}{N} \sum_{i=1}^{N} \ell_{acc}(\theta)_i$ *be the empirical risk for the data with any subset of views. Under the regularization of confidence calibration, the inequality*

$$
\widehat{R}\left( h\left( \{x_i^v\}_{v=1}^{V}; \theta \right) \right) \leq \widehat{R}\left( h\left( \{x_i^v\}_{v=1}^{V'}; \theta \right) \right)
$$

(12)

*is satisfied with* $V' \in \mathbb{N}^* \leq V$.

Next, we provide a theoretical guarantee that the conformal risk from the CMDC method is bounded by a user-defined value.

**Proposition 2 (Conformal Risk Control Guarantee).** *Given any test data point* $\{x_{test}^v\}_{v=1}^{V}$, *suppose the Conformal Risk of our model* $h\left( \{x_{test}^v\}_{v=1}^{V}; \theta \right)$ *is defined as* $CR(\theta)$, *we have:*

$$
CR(\theta) = CR_{mis}(\theta) + CR_{outlier}(\theta) \leq 2\tau.
$$

(13)

*where* $CR_{mis}(\theta)$ *indicates the risk of miscoverage that true label is not contained in the prediction set, and* $CR_{outlier}(\theta)$ *is the risk of points are wrongly identified as outliers, defined as:*

$$
CR_{mis}(\theta) = \mathbb{E} \left( \mathbb{P} \left( \{y_{test}^{true} \in \{1, \ldots, K\}\} \notin C_\theta \left( \{x_{test}^v\}_{v=1}^{V} \right) \right) \right).
$$

(14)

$$
CR_{outlier}(\theta) = \mathbb{E} \left( \mathbb{P} \left( \{\hat{y}_{test} = \{K+1\}\} \in C_\theta \left( \{x_{test}^v\}_{v=1}^{V} \right) \right) \right).
$$

(15)

*Notably, the above guarantee also ensures that:*

$$
\mathbb{P} \left( y_{test}^{true} \in C_\theta \left( \{x_{test}^v\}_{v=1}^{V} \right) \right) \geq 1 - \tau.
$$

(16)

Finally, we give a generalization analysis to demonstrate the effectiveness of our model.

**Proposition 3 (Generalization).** *Let $X$ be a set of $N$ samples with label $Y$, $\theta \in \mathbb{B}^d$ be the parameter of loss function in a finite $d$-dimensional unit ball. Define generalization risk as:*

$$
R(\theta) = \mathbb{E}_{(X,Y)} [\ell_{acc}(\theta)_i].
$$

(17)

*Let $\theta^* = \arg\max_{\theta \in \mathbb{B}^d} R(\theta)$ be the optimal parameter in the unit ball, $\widehat{\theta} = \arg\max_{\theta \in A} \widehat{R}(\theta)$ be the optimal parameter of empirical risk among a candidate set $A$. With probability at least $1 - \delta$ we have,*

$$
R(\theta^*) \leq R(\widehat{\theta}) + \frac{3 + \sqrt{2\varepsilon^2 d \ln(N) + 4\varepsilon^2 \ln(2/\delta)}}{\sqrt{N}}.
$$

(18)

Proposition 3 shows the generalization of CMDC can approach the optimal result in the order $O\left( \sqrt{d \ln(N)/N} \right)$, where $d$ indicates the number of parameters in our model and $N$ denotes the number of samples. In summary, this section demonstrates the effectiveness of our method in terms of the classification performance, conformal risk control and the generalization.

## 5 Experiments

We extensively evaluate the proposed method on real-world multi-view datasets, considering metrics such as Accuracy (ACC), Expected Calibration Error (ECE) [14], Inefficiency (Ineff), and Coverage (Cov), where Ineff measures the size of prediction sets and Cov measures the coverage of true label contained in the prediction sets.

### 5.1 Experimental Setup

*5.1.1 **Datasets**.* We conduct experiments on seven real-world multi-view datasets as follows: **HAND** [40]: This dataset consists of handwritten numerals ('0'-'9') from a collection of Dutch utility maps, the handwritten digits are represented with six different types of descriptors. **SCENE** [10]: Scene15 dataset contains 4485 images from 15 indoor and outdoor scene categories, three feature types including GIST, PHOG and LBP are extracted as multiple views. **ANIMAL** [25]: Animal consists of 10158 images from 50 classes with two types of deep feature extracted with DECAF and VGG19. **CAL** [9]: This dataset consists of 8677 images from 101 classes, where the first 10 categories are used. Deep visual features

**Table 1: Comparison with popular multi-view learning methods based on Accuracy (ACC, %). It's worth noting that our method is trained on 70% of the training data, while the other methods are trained on 80% of the training data. That's why SMDC and TMDOA outperform ours on the CUB dataset.**

| Method | HAND | SCENE | ANIMAL | CAL | CUB | HMDB | MRNet |
|---|---|---|---|---|---|---|---|
| DCCA [2] | 94.55±2.01 | 54.77±1.13 | 83.33±1.25 | 84.00±0.15 | 82.03±2.40 | 45.71±1.51 | 89.23±2.91 |
| DCCAE [42] | 97.01±0.23 | 55.12±0.23 | 85.80±0.51 | 90.03±0.11 | 85.50±1.37 | 49.12±1.00 | 88.03±0.64 |
| DTCCA [43] | 96.88±0.02 | 61.09±0.22 | 84.61±0.01 | 90.12±0.34 | 84.40±0.10 | 56.77±1.00 | 85.12±0.20 |
| CPM-Nets [47] | 94.45±1.11 | 67.09±0.05 | 86.59±0.05 | 90.05±1.42 | 89.44±0.06 | 66.84±1.21 | 88.96±0.01 |
| DUA-Nets [13] | 98.10±0.32 | 68.43±0.02 | 89.05±1.22 | 93.83±0.34 | 81.42±1.15 | 63.05±0.53 | 90.09±1.04 |
| MVTCAE [17] | 97.00±0.23 | 66.43±0.06 | 86.32±0.16 | 91.76±0.01 | 92.00±0.04 | 74.84±1.24 | 93.92±2.13 |
| TMC [15] | 98.51±0.13 | 68.30±0.01 | 89.71±0.34 | 93.41±0.22 | 91.23±1.21 | 74.98±1.02 | 92.10±1.20 |
| TMDOA [29] | 98.33±0.23 | 71.29±0.23 | 90.57±0.05 | 95.22±0.21 | 93.21±1.03 | 87.83±0.42 | 94.00±1.29 |
| ETMC [16] | 98.89±0.45 | 66.78±0.23 | 88.99±0.12 | 93.30±0.11 | 90.88±1.01 | 75.61±1.32 | 93.11±1.49 |
| SMDC [28] | 99.00±0.01 | 72.80±0.13 | 94.10±0.01 | 97.33±0.01 | **96.65±0.01** | 90.84±0.11 | 92.19±1.17 |
| CMDC (Ours) | **99.17±0.52** | **75.18±1.11** | **94.57±0.55** | **97.72±0.97** | 93.05±0.44 | **91.56±0.40** | **94.29±0.79** |

**Table 2: Ablation study 1) : Classification performance with the corresponding component.**

| Components | | | HAND | SCENE | ANIMAL | CAL | CUB | HMDB | MRNet |
|---|---|---|---|---|---|---|---|---|---|
| Fusion | $\ell_{cal}$ | $\ell_{size}$ | ACC (%)↑ | ACC (%)↑ | ACC (%)↑ | ACC (%)↑ | ACC (%)↑ | ACC (%)↑ | ACC (%)↑ |
| ✗ | ✗ | ✗ | 94.44±0.83 | 69.08±1.05 | 90.46±0.60 | 95.54±0.99 | 87.28±0.46 | 87.13±0.04 | 87.87±0.31 |
| ✓ | ✗ | ✗ | 98.33±0.46 | 74.64±0.66 | 93.94±0.47 | 97.33±1.07 | 91.74±1.31 | 90.25±1.12 | 92.70±0.21 |
| ✓ | ✓ | ✗ | 98.56±0.44 | 75.10±1.34 | 94.56±0.42 | 97.52±1.07 | 93.04±0.44 | 90.84±0.40 | 93.85±0.05 |
| ✓ | ✗ | ✓ | 98.89±0.25 | 74.67±1.24 | 94.44±0.71 | 97.43±0.59 | 93.04±1.15 | 90.41±1.17 | 93.87±0.05 |
| ✓ | ✓ | ✓ | **99.17±0.52** | **75.18±1.11** | **94.57±0.55** | **97.72±0.97** | **93.05±0.44** | **91.56±0.40** | **94.29±0.79** |

with DECAF and VGG19 are employed as two views. **CUB** [41]: Caltech-UCSD Birds dataset contains 11788 images and text descriptions from 200 categories of birds, where the first 10 categories are used. Deep features captured from GoogLeNet and text features using doc2vec are used as two views. **HMDB** [23]: This dataset is one of the largest human action recognition dataset, which consists of 6718 images of 51 categories of actions. HOG and MBH features are extracted as two views. **MRNet** [5]: This dataset includes approximately 123,330 knee Magnetic Resonance Imaging (MRI) scans for 1370 patients captured in three views, Anterior Cruciate Ligament (ACL) injury detection is used as the classification task.

*5.1.2 **Implementations***. For all datasets except MRNet, we utilize fully connected networks, while MRNet employs ResNet-18 as its backbone. We set $\tau = 0.1$, which means guaranteeing the coverage with a probability of 90%. The Adam optimizer [22] is used to train the network, where $l_2$-norm regularization is set to $1e^{-5}$. We use 5-fold cross-validation to select the learning rate from $\{1e^{-5}, 3e^{-4}, 1e^{-3}, 3e^{-3}\}$. For all multi-view datasets, the data is partitioned into training (70%), testing (20%), and calibration (10%) sets. Furthermore, we run 5 times for each method to report the average values in Figures or the mean values and standard deviations in Tables. The model is implemented by PyTorch on one GeForce RTX 4090 GPU with 24GB memory.

## 5.2 Comparison with Popular Methods

To investigate the effectiveness of our model, we conducted comparisons with several popular models for multi-view deep classification. Given the widespread use of CCA-based methods in multi-view learning, we initially compared our approach with three representative CCA-based methods: DCCA [2], DCCAE [42], DTCCA [43]. These methods employ CCA to derive latent representations, followed by classification using a support vector classifier (SVC). Next, we compared our model with three advanced multi-view representation learning approaches: CPM-Nets [47], DUA-Nets [13], and MVTCAE [17]. As in [17], logistic regression served as the basic classifier in these methods, utilizing the learned representations as input. Finally, we contrasted our model with four state-of-the-art uncertainty-based methods: TMC [15], TMDOA [29], ETMC [16], and SMDC [28]. Notably, these four methods share the same neural network architecture as ours.

The comprehensive experimental results are presented in Table 1. It is evident that our approach showcases state-of-the-art performance across all multi-view datasets except for the CUB dataset, as indicated by the ACC metric. The reason CMDC doesn't achieve the best performance on the CUB dataset is that our method utilizes a part of the training data as the calibration dataset during training. Specifically, we train our method on 70% of the training

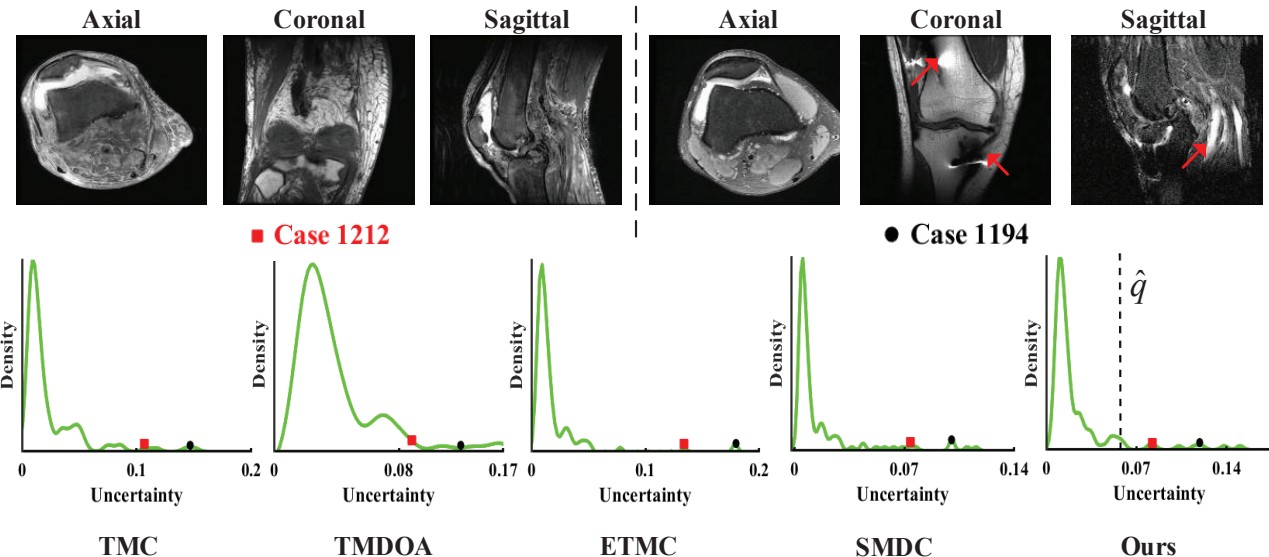

**Figure 2: In the MRNet dataset, two real-world cases with high prediction uncertainty, both labeled as ACL class ($y_{\text{true}} = ACL$), are examined. (a) The top row showcases the original images for three views (Axial, Coronal, and Sagittal). Case 1212 (left) exhibits MRI scans heavily laden with noise. Case 1194 (right) represents an abnormal sample with imperfections, such as effusion or artifacts (highlighted by the red arrows). (b) The positions of these two cases in the uncertainty density distribution of test samples are depicted across five uncertainty-aware multi-view deep classification methods. It is evident that although the estimated uncertainty of these two cases is higher compared to other test points, making a clear judgment regarding their outlier status is challenging due to the relatively small value of prediction uncertainty. In contrast, our method provides a threshold $\hat{q}$, facilitating straightforward judgments.**

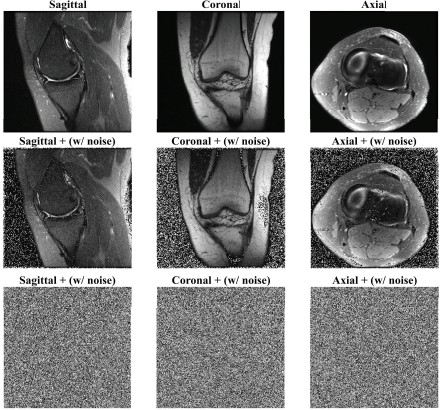

**Figure 3: Samples w/ Gaussian noise with $\eta = 10$ (middle), $\eta = 100$ (bottom) on MRNet.**

data, reserving the remaining 10% as calibration data. For fairness, the other methods are trained on 80% of the training data.

## 5.3 Ablation Study

*5.3.1 Contribution of Each Component.* We begin by assessing each key component, which includes the evaluation of the multi-view fusion strategy, the confidence calibration module, and the prediction size regularization. Apart from the results presented in

the first row of Table 2, which denote the highest accuracy among individual single-views, we conduct evaluations across all views for our primary components. As illustrated in Table 2, our method consistently surpasses all other combinations, thus confirming the efficacy of our major technical components.

**Table 3: Ablation study 2) : Evaluation of two regularizations.**

| Method | ECE ↓ | Ineff ↓ | Cov ↑ |
|---|---|---|---|
| CMDC (w/o $\ell_{cal}, \ell_{size}$) | $0.0511 \pm 0.01$ | $3.4524 \pm 0.04$ | $0.9032 \pm 0.00$ |
| CMDC (w/o $\ell_{cal}$) | $0.0357 \pm 0.01$ | $2.9963 \pm 0.11$ | $0.9041 \pm 0.02$ |
| CMDC (w/o $\ell_{size}$) | $0.0126 \pm 0.00$ | $3.2783 \pm 0.09$ | $0.9060 \pm 0.00$ |
| CMDC (full) | $\mathbf{0.0119 \pm 0.00}$ | $\mathbf{2.9901 \pm 0.02}$ | $\mathbf{0.9063 \pm 0.00}$ |

*5.3.2 Effective of Two Regularizations.* Although the inclusion of two regularization components has led to performance enhancements, their primary impact lies not in enhancing classification accuracy, but rather in enhancing calibration performance and minimizing the size of prediction sets. To evaluate these aspects, we employ three widely used metrics: Expected Calibration Error (ECE), Inefficiency (Ineff), and Coverage (Cov). We compare our method with variants that exclude either the Confidence Calibration loss ($\ell_{cal}$), the Size loss ($\ell_{size}$), or both. Quantitative results evaluated on the SCENE datasets consisting of 15 classes are presented in Table 3. The results reveal that $\ell_{cal}$ effectively diminishes ECE values

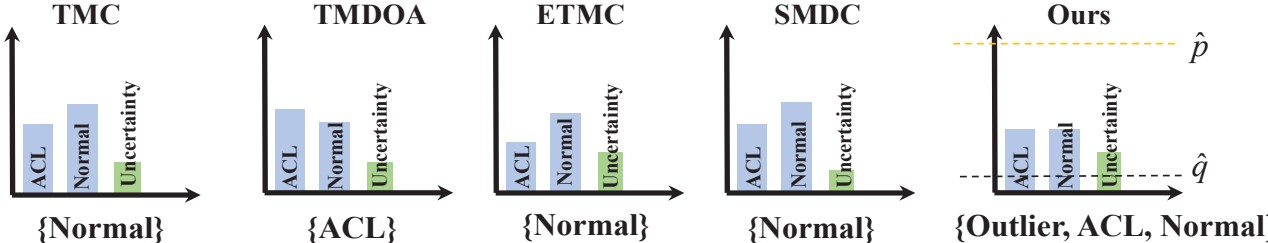

Figure 4: The class probabilities and prediction uncertainty of Case 1212 are illustrated across five uncertainty-based multi-view methods. It is observed that all methods yield nearly equal probability values between the ACL and Normal classes, largely due to the noise present in Case 1212. Even though TMDOA correctly classifies it, the judgment appears to be random. Such ambiguity poses significant clinical risk. Conversely, our method offers a prediction set containing possible classes by the thresholds $\hat{p}$ and $\hat{q}$, thereby mitigating the risk stemming from conflicts across multiple views or outlier cases.

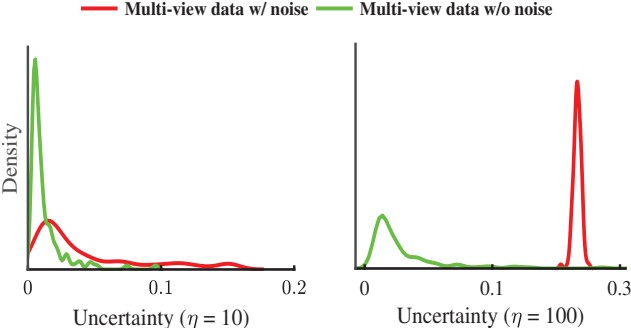

Figure 5: Density of uncertainty estimation in the MRNet dataset for the ACL task under different noise intensity $\eta$ of Gaussian noise.

across diverse multi-view datasets, signifying enhanced alignment between model confidence and actual outcomes. Furthermore, $\ell_{size}$ reduces the size of prediction sets, enhancing the reliability of results. Additionally, Cov demonstrates that our method achieves coverage with $1 - \tau = 90\%$.

## 5.4 Why We Need Conformalized Multi-view

In this section, we delve deeper into examining the practical significance of our method in real-world settings.

*5.4.1 **Uncertainty Estimation with Simulated Noises**.* Initially, we evaluate the capability of our method in estimating uncertainty. We introduce Gaussian noise $\epsilon$ to our samples, effectively rendering them as 'polluted' samples, i.e., $\widetilde{x}_i^v = x_i^v + \eta\epsilon$, where $\eta$ represents the noise intensity, and $\epsilon$ is sampled from a Gaussian distribution $\mathcal{N}(0, I)$. Figure 3 visually represents samples affected by Gaussian noise ($\eta = 100$) across the three planes in the MRNet dataset. The estimated density of uncertainty in the MRNet dataset for the ACL task is depicted in Figure 5. It can be observed that when all views are affected by high-level noise, the polluted multi-view samples can be regarded as outliers. Consequently, the overall uncertainty increases significantly compared to the clean multi-view

data, validating the good performance of our method in estimating prediction uncertainty caused by outliers.

*5.4.2 **Real-world Case Studies**.* Next, we apply our method to the real-world medical dataset MRNet to detect whether knee MRI scans of patients belong to the ACL injury category. Figure 2 showcases the prediction uncertainty density distribution estimated by five state-of-the-art uncertainty-aware multi-view deep classification methods for the imperfect Case 1194 and Case 1212 in the MRNet dataset. The experimental results from Figure 2 indicate that the uncertainty values from all five methods cannot provide clear evidence to support the doctor in making accurate diagnoses, posing a high risk due to the small difference between the estimated uncertainty values of Case 1194 and Case 1212 and those of the other samples. Our model addresses this issue by providing clear outlier class judgments using the threshold $\hat{q}$ with statistical guarantees.

Furthermore, we also present the probability maps of Case 1212 in the MRNet dataset as illustrated in Figure 4, showing the class probability and uncertainty degree across five uncertainty-based methods. It is evident that the difference between the probability of the ACL class and the Normal class is small due to the conflicts among the noisy views in Case 1212, resulting in random predictions. Our model addresses this issue by providing prediction sets using the thresholds $\hat{p}$ and $\hat{q}$ to mitigate the risks posed by inconsistencies or uncertainties from the samples, thereby enhancing the reliability of multi-view results.

## 6 Conclusion

In this study, we introduce a novel Conformalized Multi-view Deep Classification (CMDC) model aimed at generating reliable prediction sets that encompass the outlier class. This facilitates clear assessments of uncertain predictions attributed to outliers, thereby enhancing trust in decision-making. Leveraging an effective multi-view fusion strategy and conformal score function, the CMDC model provides theoretical assurances regarding the inclusion of the true label in prediction sets with a user-defined probability, while also controlling risk within a narrow margin. Empirical investigations highlight the superior performance of our approach across various multi-view datasets.

## Acknowledgments

This work was supported by the National Natural Science Foundation of China (No. 62173252), Natural Science Foundation of Shanghai (No. 21ZR1423900), and Shanghai Science and Technology Innovation Action Plan (No. 22511101903).

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
