# OpenReview forum: "Building Trust in Decision with Conformalized Multi-view Deep Classification"
_acmmm.org/ACMMM/2024/Conference — MM2024 Oral_

### Official Review · Reviewer_NMuo · 2024-05-13

**Rating:** 4
**Confidence:** 4

**Summary:**

## **1. Summary.**
This work addresses the two challenges of inaccurate single-class prediction and non-standard uncertainty measurement in multi-view classification, and combines uncertainty and anomaly categories to perform multi-view deep classification.

**Strengths:**

## **2. Strengths.**
i) This work has a reasonable motivation, a clear starting point, and comprehensive related work.

ii) Compared to previous works such as TMC or ETMC, the introduction of anomaly classes expands the boundaries of trustworthy multi-view classification.

iii) Extensive theoretical analysis and experimental verification have demonstrated the effectiveness of the proposed method.

**Limitations:**

## **3. Some minor confusion.**
After carefully reading the paper and appendix, I believe it is a carefully prepared work. However, I still have some minor confusion and suggestions regarding the theoretical analysis and other details, and I hope to discuss and reach a consensus with the authors.

**Theoretical analysis of this work:** It is worth acknowledging that the authors provided some analysis to ensure the correctness and generalization of the proposed method at the theoretical level, which is commendable. However, **1)** Regarding **Propositions 1-3**, after the authors show them in the main text, further explanations should be provided to help readers understand why the proposed proposition can be achieved, such as with **Proposition 1**, how to achieve the "**Classification Performance Improvement**"? After **Proposition 2**, how to keep the "**Conformal Risk Control Guarantee**"? Based on the **Proposition 3**, how to obtain the "**Generalization**"? More explanations can allow readers to clearly and intuitively feel the theoretical efforts of the proposed method.
**2)** A small question is, why is **Proposition 1** proven starting from $\ell_{acc}(\cdot)$? Because according to the reader's intuitive perception, the improvement of classification performance should start from $\ell_{overall}(\cdot)$ to prove the proposed statement. Similarly, why prove generalization **Proposition 3** from $\ell_{acc}(\cdot)$? Shouldn't it be the cumulative risk of global losses $\ell_{overall}(\cdot)$?
**3)**  What is the $\tau$ in **Proposition 2**? I hope the authors can further clarify to solve my confusion.

**Other small details:** **1)** Algorithms 1-3 provide a good summary of the proposed method, but it is presented from an algorithmic perspective. It is best to add a framework diagram (similar to Figure 1) to visually present the proposed method.
**2)** The theoretical analysis lacks an important part: Computational complexity, which should also be considered for inclusion in the main text to enhance the rigor of the proposed method.

**Overall, this is a good work, motivation, innovation, method presentation, and experiments are all carefully presented. And if the authors can provide effective reply on my concerns, I am willing to increase my rating.**

**Suitability:**

3

---

### Official Review · Reviewer_9BCC · 2024-05-24

**Rating:** 3
**Confidence:** 2

**Summary:**

This paper introduces conformalized multi-view deep classification and aims to address the limitations of current uncertainty-aware multi-view deep classification methods. Compared with the former research, the authors focus on set-level predictions rather than single-level ones. The proposed method minimizes the size of prediction sets while guaranteeing true labels with probability, and thus more reliable decision-making. The authors also conduct a series of experiments to validate the superiority of the proposed method.

**Strengths:**

1. The paper is easy to follow.
2. Introducing conformal multi-view deep classification into this task is indeed interesting.

**Limitations:**

1. I am curious why to fuse as eq.2. Were there any other forms to fuse the outputs?
2. In my view, multi-view classification aims to utilize multiple views with uncertainty to make the final classification more reliable and precious. According to L423-426, it seems that the correct classification or incorrect classification is quite crucial. However, how does it figure out whether the classification is correct or incorrect without labels in test scenarios?
3. In Table 1, the proposed method fails in CUB.
4. After reading the whole paper and reviewing related work, it seems that the novelty of this paper is limited. The authors introduce the conformal prediction into the multi-view deep classification task while more from intuition and lack of theoretical justification. It should be
clarified.

I am willing to increase my score if the concerns are adequately answered.

**Suitability:**

3

---

### Official Review · Reviewer_Lq85 · 2024-05-25

**Rating:** 5
**Confidence:** 4

**Summary:**

This paper introduces a novel method for multi-view deep learning, named Conformalized Multi-view Deep Classification (CMDC), which aims to enhance the reliability of decisions made from multiple views. Unlike existing uncertainty-aware multi-view deep learning methods that generate single-valued predictions with vague confidence levels, CMDC produces set-valued predictions that explicitly include an outlier category. This approach ensures that the true label is contained within the prediction set with a probability of at least 1-τ. Both empirical and theoretical analyses validate the superiority of the proposed method.

**Strengths:**

(1) The innovation presented in this paper is significant. Through an end-to-end training, CMDC effectively combines the best features of both ENN and CP approaches, offering a compelling new perspective for advancing the field of multi-view deep learning.

(2) The proposed method achieves superior performance compared to previous methods.

(3) The authors provide a comprehensive theoretical analysis to support the effectiveness of the proposed method.

**Limitations:**

(1) Figure 1 doesn't illustrate the idea of this paper but highlights an existing problem. Adding an overall framework diagram of the proposed method would aid in providing a clearer understanding of the paper.

(2) There are typographical errors present, for example, in Equation (15), $\notin$ should be replaced with $\in$.

(3) Does the size of the validation sets affect the coverage of the proposed method? A discussion regarding the impact on coverage with small-scale datasets would be interesting.

**Suitability:**

3

---

### Official Review · Reviewer_tXmc · 2024-05-27

**Rating:** 4
**Confidence:** 3

**Summary:**

This paper combines CP and EDL into the multiview classification tasks.

**Strengths:**

1. The theories are solid.
2. The experiments are extensive.

**Limitations:**

1. Why we need to introduce Conformalized Predictions into  Multiview Classification? Its not clear.
2. Comparison with ECML?
 Xu C, Si J, Guan Z, et al. Reliable Conflictive Multi-View Learning[J]. arXiv preprint arXiv:2402.16897, 2024.
3. Reliability experiments, i.e. density figures are limited to specific datasets. Can you provide more results?

**Suitability:**

2

---

### Meta-Review · Area_Chair_G3FS · 2024-06-25

**Recommendation:** Accept (Oral)
**Confidence:** 5

**Metareview:**

As all the reviewers have given positive scores, I recommend accepting this paper.